# Exposure to Antibiotics and Neurodevelopmental Disorders: Could Probiotics Modulate the Gut–Brain Axis?

**DOI:** 10.3390/antibiotics11121767

**Published:** 2022-12-07

**Authors:** Tamara Diamanti, Roberta Prete, Natalia Battista, Aldo Corsetti, Antonella De Jaco

**Affiliations:** 1Department of Biology and Biotechnologies ‘Charles Darwin’, Sapienza University of Rome, 00185 Rome, Italy; 2Department of Bioscience and Technology for Food, Agriculture and Environment, University of Teramo, 64100 Teramo, Italy

**Keywords:** autism spectrum disorders, brain development, dysbiosis, environmental factors, gastrointestinal issues, mental disorders, microbiota, neuroactive molecules, psychobiotic

## Abstract

In order to develop properly, the brain requires the intricate interconnection of genetic factors and pre-and postnatal environmental events. The gut–brain axis has recently raised considerable interest for its involvement in regulating the development and functioning of the brain. Consequently, alterations in the gut microbiota composition, due to antibiotic administration, could favor the onset of neurodevelopmental disorders. Literature data suggest that the modulation of gut microbiota is often altered in individuals affected by neurodevelopmental disorders. It has been shown in animal studies that metabolites released by an imbalanced gut–brain axis, leads to alterations in brain function and deficits in social behavior. Here, we report the potential effects of antibiotic administration, before and after birth, in relation to the risk of developing neurodevelopmental disorders. We also review the potential role of probiotics in treating gastrointestinal disorders associated with gut dysbiosis after antibiotic administration, and their possible effect in ameliorating neurodevelopmental disorder symptoms.

## 1. Introduction: Neurodevelopmental Disorders

Brain development is a dynamic process resulting from a constant interplay between genetic and environmental factors [1]. Neurodevelopmental disorders (NDDs) are a group of heterogeneous syndromes with functional impairments in the central nervous system (CNS) caused by the disruption of essential processes during neurodevelopment [2]. Brain dysfunctions during development reflect deficits in social communication, verbal, nonverbal, and social interactions and are characterized by repetitive behaviors and activities [3]. NDDs include autism spectrum disorder (ASD), intellectual disability (ID), attention deficit hyperactivity disorder (ADHD), epilepsy [4], and schizophrenia (SCZ). To date, there has been a worldwide increase in the prevalence rates of NDDs: ID, 0.63%, ADHD, 5–11%, ASD, 0.70–3%, and SCZ 0.32% [5,6,7,8,9,10]. Although NDDs tend to run in families, as suggested by twin and family studies, the inheritance patterns are complex and still under investigation [11]. Genetic alterations including de novo mutations, and rare and common copy number variants [12,13,14,15,16,17,18] have been found to be an emerging source of genetic causality [19]. Despite the huge amount of risk in genes found in the human population, there are overlapping functions that affect a limited number of biological pathways [20]. A relevant and recurrent pathway is represented by the synapse function along with CNS development. Risk genes involved in the synaptic pathway are found expressed in dendrites, axons, and pre- and postsynaptic terminals, playing different roles [21,22]. There are several genetic mutations found in synaptic genes identified in individuals, encoding for proteins including NLGNs, SHANK2, SHANK3, and CNTNAP2, providing a direct link with the synaptic function and the etiology of NDDs [23,24,25,26,27]. Although recent new technologies favor research advancements in the genetic background of NDDs, they cannot provide information about the environmental influences on genetic predispositions [28,29]. It is well established that environmental factors may alter and/or trigger childhood psychiatric conditions [30,31]. A recent rise in the prevalence of NDDs suggests a major involvement of the environment during different critical developmental windows that can be classified into prenatal, perinatal, and postnatal [32] windows. Early-life environmental injurious factors could be represented by premature birth, low birth weight, environmental contaminants, fetal exposure to smoking, alcohol, drugs, or medications during pregnancy [33,34,35]. Antibiotics are a class of drugs to which both mothers and children are exposed during pregnancy. Treatments based on antibiotics have been considered to provide minimal adverse effects on health, although epidemiologic studies have shown that early-life exposure increases the risk of obesity, asthma, and celiac disease in children [36,37,38,39]. Antibiotics have been proven to cause multiple birth defects, when administered to pregnant mothers, and are associated with several diseases in newborns by increasing the risk of congenital malformations [40,41]. Antibiotics affect the brain directly or could have an indirect effect via their actions on the human microbiome, causing long-term consequences [42,43,44].

## 2. Gut–Brain Axis

The two-way street between gut and brain has been known since ancient Grecian times, when Hippocrates affirmed that “All disease begins in the gut”, but only in 1870 was the existence of a gut–brain axis established [45]. This concept has been extended to the microbiota residing in the gastrointestinal (GI) tract, termed gut microbiota, which have been shown to be a key regulator of the gut–brain axis, and thus the microbiota–gut–brain axis has been proposed [46,47]. The relationship between gut and microbiota shows bidirectional communication with the CNS through the production of neurotransmitters, innervation via the vagus nerve, or activation of the hypothalamic–pituitary–adrenal axis [48]. Multiple factors, including mode of delivery, lifestyle, antibiotic usage, diet, age, gender, and geographic localization affect and modify the profile of the trillions of indigenous microorganisms, including bacteria, fungi, and viruses, that colonize the GI tract from birth throughout the lifespan. Additionally, gradients of pH, oxygen, antimicrobial peptides, and bile salts determine the density and diversity of microbial species along the GI tract of the host. Nowadays, it is clear that neurotransmitters such as glutamate, GABA, serotonin, and dopamine, short-chain fatty acids (SCFAs), cytokines, neuropeptides, endocannabinoids, and hormones are the molecules used for the communication between the brain and the gut through the endocrine, immune, and neural systems in order to maintain homeostatic conditions [47] in our body. Gut microbiota [48,49,50] synthesizes a vast milieu of metabolites that may directly or indirectly impact neuronal activity as well as induce host cells to produce signaling molecules, shaping both local and extraintestinal host functions [51]. Among the neuroactive molecules, SCFAs, including butyrate, acetate, and propionate, are metabolites produced from the bacterial fermentation of nondigestible fibers in the gut [52]. These play several functions in maintaining gut homeostasis, including anti-inflammatory properties. Their effects are mainly mediated by G-protein coupled receptors [53] and can also trigger regulation of gene expression and epigenetic modulation by histone deacetylases, as reported for butyrate [54].

In particular, SCFAs strongly stimulate the release of gut hormones in enteroendocrine cells [55], whereas in immune cells, selected strains might regulate T regulatory cell differentiation [56,57,58,59] and myeloid cell lineages in the bone marrow, and might affect the function of circulating mature granulocytes [59]. It has been shown that short-chain fatty acid (SCFA) levels were altered in ASD patients due to alterations in the gut microbiota composition, and administration of butyrate has been shown to improve ASD symptoms [60]. Moreover, a reduction of SCFA levels can affect both intestinal barrier and blood–brain barrier (BBB) integrity, leading to neuroinflammation with direct implications in brain functions [61]. In immunometabolism, gut microbes promote peripheral immune responses associated with CNS disorders through driving, for instance, inflammatory Th17 cell responses [62]. Long-range effects on CNS-resident immune cell function, can be promoted following the release of bacterial metabolites, such as cytokines, into the bloodstream and lymphatic systems [59]. On the other hand, the loss of intestinal barrier integrity due to gut inflammation can activate innate and adaptive immune cells to release pro-inflammatory cytokines IL-1β, IL-6, and tumor necrosis factor-α (TNFα) into the circulatory system, leading to systemic inflammation. In the last decade, the metabolism of tryptophan is gaining attention in the microbiota–gut–brain axis because this essential amino acid serves as a precursor to a variety of imperative bioactive molecules generated by both the enterochromaffin cells of the host and gut microbes [63]. Moreover, 5-hydroxytryptamine (5-HT), commonly known as serotonin, is the main product of tryptophan conversion and, incidentally, it has been demonstrated that some bacteria found in gut microbiota are able to synthesize 5-HT, affecting thus the plasma levels of this neurotransmitters [64]. It is true that 5-HT has a pivotal role in the regulation of several functions both at the intestinal and central level, such as modulation of mood, memory, and cognition [65]. It has been found that altered levels of 5-HT, mainly synthesized in the gut, are implicated by different diseases, such as irritable bowel syndrome (IBS) as well as humor disorders (i.e., anxiety and depression-like disorders) [66] and ASD [67]. During inflammation or in response to stress, tryptophan metabolism shifts to kynurenine production, leading to the formation of either kynurenic acid or quinolinic acid. The balance between these compounds is crucial in psychiatric and neurological disorders; indeed, kynurenic acid has a neuroprotective role, and quinolinic acid is involved in neurotoxicity. A variety of health benefits for the neurotransmitter of GABA have been shown, including neurostimulation and gut modulation [68,69]. The GABAergic system has an important implication also in the NDDs and neurodegenerative disorders, in terms of alterations of GABA concentrations and GABA receptor expression [70,71]. A body of evidence has associated alterations in the gut–brain axis with drug and antibiotics therapies and, in turn, altered levels of specific neuroactive molecules in different types of NDDs [60,72]. It is now clear that the gut–brain axis plays an active role in the neurodevelopmental processes, including the establishment of BBB, neurogenesis, maturation of microglia, and myelination [73] with long-term health effects (Figure 1).

## 3. NDD-Associated Gastrointestinal Symptoms

Recently, gut microbiota has assumed considerable importance for its function in influencing brain functions, including social behaviors [74]. Individuals with NDDs show different compositions of the intestinal microbiome in terms of the number and types of species. A balanced and appropriate composition of the gut microbiota confers health benefits; a disruption of this balance can reflect on brain function and behavior by acting through the microbiome–gut–brain axis [46], our “second brain” [45]. The relative abundance of the single species constituting the microbiota and its metabolites are associated with the onset of neurologic and psychiatric disorders [75]. GI disorders (ranging from severe constipation to diarrhea) are frequently observed in individuals with ASD; however, they are estimated to occur with a wide variability, from 2.2% to 96.8% of the ASD population [76]. Despite this considerable heterogeneity, overall, most studies highlight a greater prevalence of GI problems in children with ASD compared with their neurotypical counterparts, suggesting that microbial dysbiosis could contribute to gut symptoms in neurodevelopmental disorder (NDD)-affected individuals [77,78]. Moreover, a compromised gut epithelial barrier (so-called “leaky gut”) has been described in association with GI problems in NDDs [79]. This will enable bacteria and metabolites to cross the GI barrier and trigger aberrant immune responses. Accordingly, elevated levels of inflammatory cytokines in children with ASD have been reported in association with symptom severity [80]. Cytokine levels have also been correlated with ASD-associated bacterial populations (e.g., *Clostridiales*) and GI symptomatology [81,82]. Differences in microbial composition among ADHD Dutch young patients were found by sequencing 16S rRNA extracted from fecal samples [83,84,85]. A significant decrease in the gut microbial diversity in ADHD children has been reported: an unusually higher level of the family *Bacteroidaceae*, *Neisseriaceae* that causes a significant decline in the gut microbial diversity in ADHD patients, differing from a higher level of *Prevotellaceae*, *Catabacteriaceae*, and *Porphiromonadaceae* found in the control group [86]. An altered *Bifidobacterium* population during early childhood has been correlated with a high risk of developing ADHD [87,88].

Among the NDDs, SCZ is highly heterogeneous with a genetic and epigenetic component. SCZ-affected individuals are characterized by gut microbiota dysbiosis with consequential GI problems like gastroenteritis, colitis, and IBS [89,90,91,92]. Altered species diversity identifies the gut microbiota in SCZ patients [93] causing cognitive and functional anomalies [94]. However, higher levels of oropharyngeal microbial species such as *Bifidobacterium dentium*, *Lactobacillus oris*, *Veillonella atypica*, *Dialister invisus*, *Veillonella dispar*, and *Streptococcus salivarius* were found in SCZ individuals with respect to healthy controls [95,96]. In NDD individuals, there is a significant decrease or complete absence of *Bifidobacteria*, with respect to the control subjects [97]. A deficit of intestinal *Bifidobacteria* has been correlated with indigestion, vitamin B12 deficiency, dysregulated immune system, gut inflammation, depression, and anxiety-like behaviors [98,99]. The microbiota can be included among the wide range of environmental factors affecting neurodevelopment. Indeed, microbiota disruption due to antibiotics administration can impact neurogenesis and behavioral deficits [100]. Alteration in the gut microbiome composition has been observed in several NDDs conditions, including depression [101], autism [102], and other conditions [103]. Mice models have been extensively used to understand the contribution of the microbiota on behavior and how gut dysbiosis may contribute to development of NDDs. Germ-free mice colonized with fecal microbiota from children with ASD show increased autistic-like behaviors in comparison to controls [104]. Moreover, specific bacteria groups, such as *Clostridiaceae*, *Lactobacillales*, *Enterobacteriaceae*, and *Bacteroides*, were found to be enriched in the ASD gut microbiota composition in mice in comparison with the control microbiota. Interestingly, mice that were treated with gut microbiota from ASD patients displayed an alternative splicing in the brain of ASD-related genes [105]. Several mutations have been reported in risk genes for NDDs, genetic mouse models have been generated by introducing the human mutation in the endogenous murine gene, whereas environmental models of ASDs are characterized by the exposure to a specific external influence [106]. Both types of mice models have been considered for studying how the genetic or environmental modification impacts on the gut–brain axis [107]. The R451C mutation, mapping in the postsynaptic protein neuroligin3 (NLGN3), was found in two affected brothers [108] who, beyond their behavioral deficits, displayed GI dysfunction including chronic gut pain, diarrhea, and esophageal regurgitation [109]. The mouse model, expressing the human R451C NLGN3 mutation, displays in vivo small intestinal motility and increased numbers of myenteric neurons in the small intestine, suggesting that the mutation alters enteric nervous system (ENS) function and structure [109]. In addition, R451C NLGN3 mutation alters mucus density and the spatial distribution of bacteria species in the GI tract of mice [110]. Additionally, in the knockout (KO) mouse model for NLGN3, gut dysfunction is characterized by a faster colonic motility and an increased colonic diameter, although GI structure and enteric neuron populations were unaltered [111]. Similarly to the alterations observed in mice models for NLGN3, the deletion of *Shank3*, a leading ASD candidate gene encoding for a neuronal scaffold protein implicated in the organization of the synapse through several protein–protein interactions among which are neuroligin-1, alters gut function and the microbiome [112,113]. SHANK3αβ KO mice show altered GI morphology and display differences in the composition of fecal microbiota [112]. Mutations in chromodomain helicase DNA binding protein 8 (*CHD8*) are among the most common de novo mutations associated with ASD in the developing brain [114]. Mutations in *CHD8* increase susceptibility to GI issues in affected individuals [115,116]. *CHD8* mutations associated with ASDs caused lower intestinal motility when expressed in zebrafish [115]. The CHD8+/−mouse model has a shorter intestine [117], the width of the mucus layer is lower in the small intestine of CHD8+/− mouse, and the number of goblet cells is reduced. Moreover, it was found that the CHD8+/− mouse has a higher bacterial load and microbiota diversity in the colon tract with respect to WT mice [118]. Fragile X syndrome is an NDD caused by a mutation in the X-linked *FMR1* gene [119]. In the *Fmr1* KO2 mouse, the gut microbiome is altered and associated with different bacterial species population, belonging to genera *Akkermansia*, *Sutterella*, *Allobaculum*, *Bifidobacterium*, *Odoribacter*, *Turicibacter*, *Flexispira*, *Bacteroides*, and *Oscillospira* [120]. A nongenetic, idiopathic ASD mouse model is the black and tan brachyury mouse strain (BTBR), which is characterized by repetitive behavior and social deficits [121]. The BTBR mouse model displays a marked intestinal dysbiosis with respect to the WT strain. The GI profile exhibits an altered gut microbial composition, altered social behavior, increased gut permeability, and colon proinflammatory biomarkers [122]. During pregnancy, alterations of the maternal gut can influence the microbial diversity and immunity in the offspring, predisposing them to the onset of NDD conditions [123,124,125]. This is the case of the maternal immune activation (MIA) mouse model, in which pregnant mice are administered with potent immune activator and generate pups with ASD-like behaviors [126]. MIA mice show decreased intestinal barrier integrity, dysbiosis of the intestinal microbiota and neurodevelopmental abnormalities in the offspring [127]. MIA offspring display an altered serum metabolomic profile, increased gut permeability, and abnormal intestinal cytokine profiles, such as the IL-6 [128], IL-1β, and TNF-α and also the total number of bacteria is significantly reduced in MIA offspring [129].

The influence of the microbiota composition and its effects on gut–brain communication is not fully clear and involves multiple mechanisms and factors. The microbiota colonization is a pivotal event; the gut microbiota changes during pregnancy, and maternal antibiotic administration during lactation influences the milk composition, which can affect the infant gut microbial composition [130]. In other words, the human gut microbiota exhibits distinct and singular metabolic traits characterized by a maternal signature [131]. At present, it is not possible to define a single bacterium as a hallmark of NDDs despite the significant increase, decrease, or complete absence of species found in affected individuals and mouse models of disorders. The increased or decreased abundance of gut microbes can be correlated with disrupted GI mucosal barrier, pathological intestinal conditions, and decreased immune surveillance, due to altered gut metabolite production. It is important to underline that some bacteria species have positive or negative effects on the gut–brain axis outcomes in humans and in mouse models and the same components of the gut microbiota do not have the same effect on different individuals, as suggested by the evidence obtained in both genetic and environmental NDD mouse models; in fact, single bacteria species may be important either for health or disease conditions.

## 4. Prenatal Antibiotics Exposure and the Risk of ASD

Pregnancy is a crucial period for fetal brain development. Embryonic brain development can be divided into three main stages: the first trimester, which is characterized by the formation of the neural tube and the production of neural progenitor cells and neurons; the second trimester, when neurons migrate to the cortical layer and begin to form synaptic connections; and finally, the third trimester, during which neuronal axons, glia and oligodendrocytes, are integrated into neural circuits [132]. During the prenatal period, the fetal brain is particularly sensitive to the surrounding environmental stimuli impacting on the CNS development [133]. A possible disturbance of the prenatal environment due to drug exposure, such as antibiotic administration, might cause neurodevelopmental alterations and subsequently lead to the onset of NDDs. Antibiotic treatment during pregnancy is continuously increasing as intrapartum prophylaxis; however in utero exposure to treatments may affect the newborn [134,135,136]. The most frequently prescribed antibiotics are “macrolides’’ that include erythromycin, azithromycin, and clarithromycin [137]. A systematic study on the use of macrolides, during the first trimester of pregnancy, showed consistent evidence of an increased risk of abortion and cardiovascular fetus malformation [40]. A large epidemiological work associates maternal infection and the use of antibiotics during pregnancy with an increased risk of developing NDDs [138] and altered brain functions in the offspring. However, it is still unclear how the exposure to antibiotics in utero affects the maturation of the gut microbiome from the fetal to the adult and the development of CNS in children. A long-term follow-up in children whose mothers took part in the “ORACLE II’’ trial of antibiotics, showed an increased prevalence of cerebral palsy associated with antibiotic treatment [138]. Neural tube defects were reported more frequently in children from women exposed, for 12 weeks before conception, to trimethoprim, an antibiotic that blocks the dihydrofolate reductase enzyme responsible for the conversion to folate [139], which is necessary for the closure of the neural tube during development. This is indicating a direct correlation between the treatment of the mothers and the newborn. Longer treatments increased the correlation between antibiotics and ADHD development [140]. Population-based studies showed an association between the maternal use of antibiotics during pregnancy and ASD in children [141,142]. Two main studies involved 96,736 and 780,547 children, respectively. The first study reported a significant increase in the risk of ASD after treatment with antibiotics during pregnancy [142]. In the second one, a strong association between prenatal exposure to antibiotics was correlated with the risk of ASD [141]. The associations between antibiotic exposure and later development of NDDs could reflect the direct effects on the gut–brain axis. Prenatal exposure was associated with a 32–41% (hazard risk ratios (HR) = 1.32–1.41) increased the possibility of sleep disorders, whereas the risk estimates for NDDs increased from 12% to 53%. The analysis has included antibiotics used against airway, urinary tract, skin and, soft tissue infections [143].

To deeply explore the contribution of maternal gut microbiota during pregnancy on newborn brain development, researchers have used animal models treated with antibiotics known to alter the maternal GI tract. Tochitani et al. showed that the administration of antibiotics to pregnant dams, during the embryonic stage (E9–E16), perturbs the maternal composition of the gut microbiota and flora in the offspring at the stage of postnatal period (P24) and this affects social behavior and locomotor activity with respect to control animals [144]. Similar evidence comes from a recent study where C57BL/6J mouse dams were exposed to antibiotic treatment dissolved in drinking water from gestational day 12 through offspring of the postnatal period (P14). Male and female offspring display ASD-like behaviors, including alteration in ultrasonic vocalization production during maternal separation and altered offspring thermoregulation in comparison to age-matched control [145].

During pregnancy, the role of the gut perturbation, due to antibiotic administration, on brain development and CNS dysfunction has been studied on rat models. Females were exposed, during the gestational period, to antibiotics that proved to alter the social behavior of the offspring [146]. In the same study, pups exposed to prenatal antibiotic treatment displayed anxiety-like behavior and greatly reduced social interactions [146]. Voung and colleagues identify that early mid-gestation is a critical period during which the maternal microbiome promotes fetal thalamocortical axonogenesis in the offspring in order to support developmental processes regulating behaviors in adult mice [147]. Taken together, these findings support the influence of maternal stimuli on fetal development; however, the molecular pathways implicated, and the metabolites involved, still remain unclear.

## 5. Early Antibiotics Exposure and the Risk of ASD

Brain plasticity is the change in neuronal networks in response to various stimuli that can permanently shape the brain [148]. Neuronal plasticity is sensitive to internal and/or external stimuli, and the interaction between genes and the environment are influenced by a variety of factors [149]. Childhood and adolescence are pivotal periods for brain development which include critical events such as neurogenesis, axonal dendritic growth, and synaptogenesis [150]. Antibiotics, often essential for treating infections in an early stage of life, can promote long-lasting adverse effects on brain development [151,152,153,154,155]. The use of antibiotics in children who develop NDDs was shown by several studies [156,157,158]. A higher risk of developing severe mental disorders at an adult age was found in children and adolescents treated with antibiotics, with the most pronounced effects observed by the use of a broad and moderate spectrum of antibiotics [159,160]. Early-life antibiotic exposure has been linked to a lower intelligence quotient and social scores, as well as higher behavioral difficulty scores, suggesting that it may represent a risk factor for ADHD, depression, and anxiety disorders [161]. ADHD is one the most common NDDs, and several studies have demonstrated that the exposure to antibiotics in early life alters the equilibrium in the gut microbiota and contributes to the development of ADHD [162]. The window between 0 and 2 years represents an important period in brain development, wherein changes, modification, and organization of the brain occur more and more rapidly than at any time during childhood [163]. The relationship between antibiotic exposure, in the first 2 years of life, and cognitive deficits was examined in a cohort of 342 children at the age of 11. This study showed that toddlers treated with antibiotics had an increased risk of developing behavioral deficits and depression symptoms during childhood [164]. The association between the use of antibiotics during early life, ADHD, and ASD was also studied in twins from 7 to 12 years old in the Netherlands and in 9-year-old twins from Sweden. In both studies, children that were temporarily exposed to antibiotics between 0 and 2 years (any pharmaceutical formulation, oral or intravenous, defined as parent-reported) resulted in an increased risk of ADHD and ASD; however the importance of the familial environment and the genetics influence in the etiology of NDDs has also to be considered [165]. Children exposed in the first two years of life to the most prescribed antibiotic classes (penicillins, cephalosporins, and macrolides that markedly impact on microbiota composition (see Table 1)) were more likely to develop asthma and allergic rhinitis and atopic dermatitis and ADHD [36]. The effect of postnatal exposure to penicillin was tested on a cohort of 677,403 children that resulted in an increased risk of developing ASD in comparison to untreated children [166]. Children are more susceptible to bacterial infections than adults, and severe infections regarding the nervous system during childhood might also result in the onset of NDDs later in life [167,168,169,170,171]. Infections treated with antibiotic drugs, and infections requiring hospitalizations in particular, were associated with increased risks of SCZ and psychiatric disorders, which may be mediated by effects of infections/inflammation in the brain, alterations of the microbiome, genetics, or other environmental factors. The connection between infections during periods of rapid growth, brain plasticity and diagnosis of NDDs may be explained by microbial metabolites, produced by the enteric bacteria, interfering with normal brain development. During normal conditions, gut metabolites produced by *Lactobacillus* and *Bifidobacterium* spp. produce the inhibitory neurotransmitter GABA affecting its activity in the brain. Antibiotic treatment alters the production of metabolites, such as SCFAs or amino acids, that may lead to dysfunction of the epithelial barrier in the intestine and BBB in the brain. Individuals with NDDs were more likely to have experienced severe infections of CNS from age 0–3 years [172]. A study on a Danish population confirms these observations: a wide range of NDDs in relation to previous CNS infections are reported and a significant association (HR 3.29) with developing ID and ASD [173]. Otitis media is one of the most common infections in childhood with high incidence in children with ASD [174]. Wimberley and colleagues found an increased case of ASDs associated with otitis media infection and antibiotics treatment in a study based on a Danish cohort of 780,547 children [141]. The correlation between infections and antibiotic treatment may contribute to a later diagnosis of NDDs [167]. Antibiotics used to treat gastroenteritis caused by Shigella infection, when administered at a young age (5–18 years), increase risks of developing ADHD respective to children who did not [175], confirming that the antibiotic treatment by affecting the human microbiome, plays an important role in developing NDDs [176].

The effects of early exposure to antibiotics on gene expression and behaviors has been widely documented in several in vivo studies using the murine model [177,178,179,180]. Leclercq and colleagues exposed mice of both sexes to a low dose of penicillin in late pregnancy and early postnatal life and showed changes in anxiety-like and aggressive behavior and decreased sociability [181]. They showed antibiotics treatment caused long-lasting changes in mice gut microbiota and altered BBB integrity in the hippocampus brain region [181]. At the molecular level, antibiotics administration altered the expression of molecules involved in memory and learning like the brain-derived neurotrophic factor (BDNF) that is crucial for promoting neuronal survival and synaptic plasticity [182]. In young mice, disruption of gut microbiota by antibiotics, shows deficits in memory retention and leads to a significant reduction of BDNF production in the hippocampus of the adult brain [183]. Levels of BDNF and its receptor, tropomyosin-related kinase B, are downregulated in the hippocampus and are unchanged in the prefrontal cortex of treated animals [182]. Dysbiosis in mice can be obtained by exposing young animals to a cocktail of a broad spectrum of antibiotics that cause gut inflammation, depressive-like symptoms, social behavior and cognitive deficits, along with changes in brain neuronal firing and microglial–glial activation [178]. A recent study shows that perinatal exposure to antibiotics affects cortical development with a long-lasting effect on brain functions in young mice [184]. Perinatal penicillin exposure significantly increased sensorimotor gating and decreased the ability to discriminate between textures in adolescent mice. These behavioral alterations were accompanied by increased spontaneous neuronal activities and a delayed maturation of inhibitory neuronal circuits [184]. An excess use of antibiotics induced neurotoxic effects on mice brain [185]. Amoxicillin administration, at clinical doses, has been reported to induce depression in young rats [186]. Significant changes in gene expression have been observed in both the frontal cortex and amygdala after 10 days exposure to a low dose of penicillin to postnatal mice [187]. Alterations in the microbiota were more extensive in mice that were exposed to antibiotics during the gestation of the dams, confirming the connection and the transfer between the maternal microbiota and the embryos with respect to mice treated after birth [187]. These results provide evidence that early-life antibiotic exposure in humans and mice have effects not only on the gut microbiome but also on gene expression within critical brain structures, which are vulnerable to perinatal insults [188]. Early-life antibiotic exposure causes unexpected consequences on childhood health; however, these findings require further validation.

## 6. Probiotics and Use of Probiotics in NDDs

In this paragraph, all the names of lactobacilli are cited according to the original species names, preceding the reclassification of *Lactobacillus* genus that have published in 2020 [191]; for the current nomenclature please see the following link: http://lactotax.embl.de/wuyts/lactotax/ (last updated on 2 September 2021)). According to the current definition, probiotic bacteria, traditionally belonging to Gram-positive taxa, (i.e., Lactobacilli and Bifidobacteria), are “live microorganisms that, when administered in adequate amounts, as part of a food or a supplement, confer a health benefit on the host” [192]. In the last several decades, probiotics have been largely used as adjuvant in the treatment of several diseases, mainly for the maintenance of a healthy gut environment through their impact on gut microbiota composition, interactions with the intestinal epithelium, and finally with the immune system [193]. So far, the role of probiotics in treating GI disorders is well known, especially for the restoration of gut dysbiosis associated with antibiotic administration, but probiotics have shown the potential to have a broad spectrum of health benefits, ranging from digestive to neurodevelopment and neurodegenerative disorders [65]. In this context, due to the crucial role of the gut microbiota in modulating human brain function via the gut–brain axis [194], a particular class of probiotics, defined with the term “psychobiotic” have shown the ability to specifically confer health benefits at the brain level. As conventional probiotics, psychobiotics can directly modulate the gut microbiota composition and functionality. During their transient colonization at the intestinal level, they can contribute to the maintenance of a healthy gut microbiota by producing growth factors that favor the growth of beneficial microbes, for example during antibiotic therapy, by competing for nutrients and/or producing inhibiting molecules that protect from pathogens colonization, and by interacting with the intestinal mucosa and modulating the intestinal immune system [70]. The most speculated mechanism of action by which psychobiotics exert their beneficial effects for mental health is the production and/or stimulation of different types of neuroactive molecules, previously described, directly involved in the two-way microbiota–brain communication [195]. Lactobacilli and Bifidobacteria are involved in the production of GABA with some *Lactobacillus* spp. and also in the production of acetylcholine, whereas *Bacillus* species can stimulate the production of dopamine and noradrenaline. Serotonin has been found to be produced by certain *Escherichia*, *Enterococcus*, and *Streptococcus* species [195,196]. Moreover, the production of neurotransmitters has been reported as a species-specific feature in *Lactobacillus* genus [197]. Intestinal bacteria can also be involved in modulating neurotransmitter levels by regulating the metabolism of neurotransmitter precursors, as the case of increased plasma tryptophan levels by *Bifidobacterium infantis* [198] or, for example, by indirectly stimulating the serotonergic system through the production of SCFAs [199]. Interestingly, bacteria from food origins have been recently shown to be able to modulate neurotransmitters [200]. For example, *Lactobacillus plantarum* DR7 improved stress, anxiety, and cognitive functions by stimulating dopamine and serotonin pathways [201,202], whereas the food-associated *L. plantarum* C29 showed the ability to improve cognitive functions in adults with mild cognitive impairments [203]. Several studies reported the ability of different probiotics species in restoring neurotransmitter levels in diverse neurological diseases. In Table 2, we summarize several pieces of preclinical and clinical evidence of the use of probiotics in NDDs, and we discuss this evidence below. *Lactobacillus rhamnosus* (JB-1) has been found to reduce stress-induced corticosterone and anxiety- and depression-related behavior in mice by modulating GABA expression in the brain via the vagus nerve [204]. Similar effects have been reported for *Lactobacillus helveticus* R0052 and *B. longum* R0175 effects in rats [205]. Amelioration of depression-like behavior via reduction of 5-HT and dopamine levels in the brain of rats have been found after administration of *Bifidobacterium infantis* 35624 [206]. Intake of *Bacteroides fragilis* restores normal 5-HT levels in an ASD animal model [207], whereas *L. plantarum* of PS128 increased the dopamine level in the prefrontal cortex in early-life stress mice [208] and improved many of the behavioral aspects of ASD, such as disruptive and rule-breaking attitudes and hyperactivity/impulsivity in children [208]. A strain of *L. plantarum* (MTCC1325) was able to restore acetylcholine also in rats affected by neurodegenerative disorders [209]. Probiotic *Lactobacillus helveticus* NS8 also showed the ability to modulate neurotransmitters, such as BDNF, serotonin, and noradrenaline in the hippocampus of rats as well as to increase circulating antiinflammatory cytokines, leading to improvement of both the intestinal barrier and the BBB, and in turn ameliorating the global inflammation status [210]. Although the entire mechanism by which probiotics ameliorate diverse NDDs symptoms is still unrevealed, a healthy gut microbiota, and in turn, the maintenance of a proper signaling network from ENS to CNS is recognized to be fundamental for proper brain functions; thus, the use of probiotics and related metabolites as an alternative intervention strategy to ameliorate and/or counteract NDDs is emerging (and clinical trials are increasing but still limited). The generation and the use of ASD animal models has shown not only that the microbiota is essential for development of social behavior [211], but also that restoring normal gut microbiota components with probiotics can correct GI permeability defects. In fact, the altered microbial composition and ASD-related abnormalities are linked to reduced intestinal production and toxin absorption [128]. Probiotic administration has been shown to improve ASD symptoms [212], and to prevent somatic symptoms in SCZ [213], and in drug-resistant epilepsy [214]. Intake of fermented food and probiotics, such as *Bifidobacterium* spp. and *Lactobacillus* spp., have been shown to ameliorate psychiatric disorder-related behaviors, including anxiety, depression, obsessive-compulsive disorder, and memory skills, as well as to attenuate stress responses [215]. Two randomized control studies by Santocchi et al. [216,217] evaluated the effects of a diet supplemented with a mix of probiotics, called De Simone Formula (labeled as Vivomixx^â^ in EU) on the main symptoms of ASD in preschool children with and without GI symptoms. The treatment, which involved the administration of eight probiotic strains, has shown significant effects not only in the improvement of GI symptoms but also in multisensory processing and adaptive functions [217]. Recently, Kalenik et al. reported some randomized trials in which different probiotic strains have been administered in ADHD children to evaluate the effect of probiotic supplementation with the occurrence of ADHD symptoms [218], but 4 out of 5 studies have been shown no substantial differences in cognitive and neurodevelopmental outcomes. However, in one study evaluating the impact of early administration of *Lactobacillus rhamnosus* GG on the development of ADHD in infants, probiotic supplementation showed a preventive effect in reducing the risk of developing ADHD [87]. Interestingly, in 2019 a promising multinational clinical trial has been started to evaluate the effect of the administration of a symbiotic formula (Synbiotic 2000 Forte 400) containing a mixture of different probiotics (*Pediococcus pentosaceus*, *Lactobacillus paracasei* subsp. *Paracasei*, *L. plantarum*, and *Leuconostoc mesenteroides*) and four prebiotics (β-glucan, inulin, pectin, and resistant starch) in a cohort of 180 adults with ADHD [219]. To date, clinical studies applying probiotics in ADHD patients are still limited and future studies are needed to achieve sufficient evidence to recommend probiotics administration as beneficial treatment in ADHD. Probiotics supplementation has been applied as an alternative treatment in SCZ, even though the literature is still limited, and more clinical studies are needed. A systematic review by Ng et al., associates the effect of probiotic administration with the amelioration of side-effect symptoms of SCZ, mainly related to the antipsychotic therapy, such as perturbation of the microbiota composition that may lead to adverse metabolic effects, weight gain, constipation, and finally to systemic inflammation and neuroinflammation [220]. *Lactobacillus rhamnosus* GG and *Bifidobacterium animalis* subsp. *lactis* strain Bb12 have shown to have positive effects in improving intestinal barrier integrity and ameliorating bowel difficulties in patients with SCZ via modulation of inflammatory cytokines belonging to IL-17 family, but no significant impact on positive and negative syndrome scale (PANSS) psychiatric symptom scores [221]. Improvements in constipation and insulin resistance have been found by Nagamine et al. [222] after four weeks of treatment with BIO-THREE^®^, a mixture of *Streptococcus faecalis*, *Bacillus mesentericus*, and *Clostridium butyricum*, in SCZ patients, whereas similar results with no changes in PANSS were reported also by other clinical studies using the same mixture of probiotics (*L. rhamnosus* GG and *B. animalis* subsp. *lactis* strain Bb12) in patients with psychotic symptoms [213,223]. However, Okubo et al. (2019) reported that the probiotic strain *Bifidobacterium breve A-1* was able to improve PANSS scores, depression, and anxiety in SCZ patients after four weeks of treatment. Moreover, the authors correlated the amelioration of symptoms with a modulation of IL-22 and TNF-related activation induced cytokines (TRANCE), involved in the intestinal barrier functions [224]. A combination of probiotic mixture (*Bifidobacterium bifidum*, *Lactobacillus acidophilus*, *Lactobacillus fermentum*, and *Lactobacillus reuteri*) and vitamin D have been administered in SCZ patients, showing amelioration of PANSS scores with reduced inflammation and enhanced plasma total antioxidant capacity [225]. Munawar et al. [226] extensively reviewed diverse nonpharmaceutical approaches in the treatment of SCZ, including probiotics and prebiotics, suggesting the use of a psychobiotic in SCZ as a promising challenge for clinical research.

Preclinical studies and human trials show that probiotics, mainly psychobiotics, can be beneficial for brain health and thereby they could be a promising alternative therapy for the treatment of NDDs. However, some aspects need to be considered and more deeply elucidated, such as the strain specificity, the probiotic dosage, time of treatment, and the precise mechanism of action at the molecular level. Moreover, there are some limitations, including individual differences (i.e., genetic background, environmental factors, diet, gender) and/or the low number of participants which remains a limit in producing high-quality clinical data [53].

## 7. Conclusions

Prenatal, natal, and postnatal adverse factors represent the underlying conditions for the onset of NDDs during brain development. NDDs are multifactorial disorders involving both a strong genetic component and environmental contributors. This wide and complex group of factors makes it difficult to find a trigger. The genetic and phenotypic complexity underlying NDDs are still the main obstacle to finding effective therapies. We have focused on the effects caused by an environmental factor, represented by antibiotic administration during different stages of CNS development. Antibiotic administration has been proposed as a possible therapy for ASD patients; however, this treatment can have side effects by affecting the gut microbiota homeostasis by targeting both pathogens and healthy commensal bacteria. It is now accepted that antibiotics alone or added to a genetic risk may perturb the gut–brain axis and have effects on the correct development of the brain. To date, mounting evidence from human and animal studies suggests that gut microbial targeting therapy may be beneficial as a new and safe method for treating individuals affected by NDDs. Microbiota is indeed influenced by different environmental factors before birth, during infancy, and during childhood, and can play a key role CNS development, influencing neurogenesis and microglial maturation. Alterations in the gut microbiome caused by antibiotics administration in children can lead to inappropriate neuronal maturation during critical phases of brain development. On the other hand, a particular class of probiotics, defined as “psychobiotics”, can specifically confer health benefits at the brain level. They can transiently colonize the gut, and in turn, restore the composition of a healthy gut microbiota by producing growth factors for beneficial microbes, for example during antibiotic therapy, by competing for nutrients and/or producing inhibiting molecules that protect from pathogens colonization, and by interacting with the intestinal mucosa and modulating the intestinal immune system as well as by producing and/or stimulating different types of neuroactive molecules directly involved in the two-way microbiota-brain interplay. In summary, evidence from preclinical and clinical studies provides support for the promising effect caused by probiotic administration. The future holds the exciting potential of probiotic-based therapies to prevent and cure the onset of NDDs.

## Figures and Tables

**Figure 1 antibiotics-11-01767-f001:**
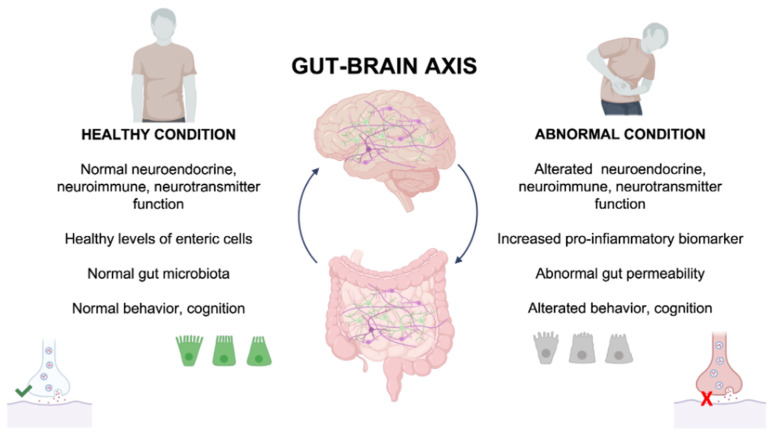
The gut–brain axis and the main effects during healthy and pathological conditions.

**Table 1 antibiotics-11-01767-t001:** Effect of 3 antibiotic classes on human gut bacteria levels [189,190].

Antibiotics	Human Gut Bacteria	
Penicillin (Amoxicillin, ampicillin, Oxacillin, PenV)	↑Enterobacteria ↑*Bacteroidaceae*	↓Bifidobacteria ↓Lactobacilli ↓Eubacteria
Cephalosporins (Cefalor, Cafotaxime, Cefuroxime, Cefepime)	↑Clostridia ↑*Bacteroides* spp.	↓*E. coli* ↓Bifidobacteria ↓*Enterobacteriaceae*
Macrolides (Azithromycin, Clarithromycin, Erythromycin, Spiramycin)	↑Enterococci ↑Streptococci ↑*Bacteroidetes* ↑Enterobacteria	↓Actinobacteria ↓*Clostriales* spp. ↓*Veillonella* spp.

↓decrease; ↑increase

**Table 2 antibiotics-11-01767-t002:** Preclinical and clinical evidence of the use of probiotics in NDDs.

Bacteria	Health Condition	Experimental Model	Main Outcomes	Reference
*L. rhamnosus* JB-1	Healthy condition	Adult male BALB/c mice	↓stress-induced corticosterone and anxiety- and depression-related behavior Modulation of GABA expression at the brain level via the vagus nerve	[227]
*L. plantarum* DR7	Mental stress condition	Adult patients	↓stress; ↓anxiety; ↑cognitive functions ↑dopamine and norepinephrine ↓plasma cortisol and pro- inflammatory cytokines	[201]
*L. plantarum* DR7	Mental stress condition	Adult patients	↓stress; ↓anxiety; ↑cognitive functions Modulation of stress-induced bowel movement and gut microbiota in association with dopamine and serotonin	[202]
*L. plantarum* P8	Mental stress condition	Adult patients	↓stress; ↓anxiety; ↑memory and cognitive traits ↓pro-inflammatory markers	[228]
*L. plantarum* MTCC1325	Neurodegenerative disorders	Albino rats	Behavioral changes; ↓cognitive deficits ↓acetylcholine levels	[209]
*L. plantarum* C29	Adult with mild cognitive impairments	Adult patients	↑cognitive functions especially in the attention domain ↑serum BDNF levels	[203]
*L. casei* Shirota	Chronic fatigue syndrome	Adult patients	↓anxiety symptoms with modulation of gut microbiota	[229]
*B. infantis* 35624	Maternal separation (MS) model	MS adult rat offsprings	Normalization of the immune response; ↓behavioral deficits ↓noradrenaline in the brain ↓depression-like behavior; ↓5-HT, noradrenaline and dopamine levels; ↑peripheral IL6	[206]
*L. helveticus* R0052 combined with *B. longum* R0175	Induced stress	Adult Wistar rats	↓anxiety-like behavior; ↓stress-induced gastrointestinal discomfort	[205]
*L. helveticus* R0052 combined with *B. longum* R0175	Induced stress	Adult patients	↓psychological distress; ↓stress-induced gastrointestinal discomfort	[205]
*L. helveticus* NS8	Chronic restraint induced stress	Sprague-Dawley rats	↑intestinal barrier and BBB; ↓global inflammation status ↑BDNF; ↓serotonin and noradrenaline in the hippocampus ↑circulating anti-inflammatory cytokines	[210]
*L. plantarum* PS128	Early life stress condition	Adult C57BL/6J mice	↓locomotor activities anxiety-like and depression-like behaviors ↓serum corticosterone and inflammatory cytokine levels ↑anti-inflammatory cytokine levels ↑dopamine and serotonin in the prefrontal cortex	[208]
*L. plantarum* PS128	ASD	Children	↓Age-dependent autism symptoms; ↓behavioral aspects, such as disruptive and rule-breaking attitudes and hyperactivity/impulsivity	[230]
*Bacteroides fragilis*	ASD	Maternal immune activation murine model	↓ASD-related defects in communicative, stereotypic, anxiety-like and sensorimotor behaviors Restoration of gut permeability and modulation of gut microbial composition	[128]
Vivomixx^→^ VSL#3	ASD	12-year-old boy	↓severity of abdominal symptoms; ↓Autistic core symptoms Modulation of gut microbiota and positive regulation of intestinal barrier	[212]
Vivomixx^→^ VSL#3	ASD	Preschool children	↓GI symptoms; ↑multisensory processing, adaptive functions, and developmental pathways	[216,217]
*L. acidophilus* DSM32241, *L. plantarum* DSM32244, *L. casei* DSM32243, *L. helveticus* DSM32242, *L. brevis* DSM11988, *B. lactis* DSM32246, *B. lactis* DSM32247, and *S. salivarius* subsp. *thermophilus* DSM32245.	Drug-resistant Epilepsy	Adult patients	↓epileptic seizures; ↑quality of life ↓serum IL-6 and sCD14; ↑serum GABA	[214]
*L. rhamnosus* GG	ADHD	Infants	Preventive effect in reducing the risk of developing ADHD	[87]
*L. rhamnosus* GG combined with *B. animalis* subsp. *lactis* Bb12	SCZ	Adult patients	↓severe bowel difficulty and prevention of common somatic symptoms associated with SCZ Positive effects on digestion and on GI disorders such as chronic constipation	[213]
*L. rhamnosus* GG combined with *B. animalis* subsp. *lactis* Bb12	SCZ	Adult patients	↑intestinal barrier integrity; ↓bowel difficulties via modulation of inflammatory cytokines belonging to IL17 family no significant impact on positive and negative syndrome scale (PANSS) psychiatric symptom scores	[221]
BIO-THREE^®^	SCZ	Adult patients	↓constipation; ↓insulin resistance; ↓intestinal inflammation	[222]
*B. breve* A-1	SCZ	Adult patients	Improved PANSS scores; ↓depression; ↓anxiety ↓symptoms with a modulation of IL22 and tumor necrosis factor-related activation induced cytokines (TRANCE)	[229]
*B. bifidum*, *L. acidophilus*, *L. fermentum* and *L. reuteri* combined with vitamin D	SCZ	Adult patients	Amelioration of PANSS scores ↓inflammation; ↑plasma total antioxidant capacity	[225]

↓decrease; ↑increase

## Data Availability

Not applicable.

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
