# Peer review of "Exposure to Antibiotics and Neurodevelopmental Disorders: Could Probiotics Modulate the Gut–Brain Axis?"

_antibiotics, 2022, doi:10.3390/antibiotics11121767_

Round 1

Reviewer 2 Report

The manuscript from Diamanti et al., is a review of the contribution of dysbiosis to neurodevelopmental disorders (NDDs) and the potential of probiotics as therapeutic. The manuscript reads more like an essay rather than a structured review article. The authors introduce NDDs and the gut-brain axis in sections 1 & 2. In section 3, they describe the association between dysbiosis and neurodevelopmental disorders. The authors then detail the effect of antibiotic use and its contribution to the development of NDD in sections 4 & 5, followed by the effect of probiotics in section 6. Overall, the manuscript provides a detailed description of the role of gut microbiota in NDDs. I have two comments:

1. Section 4- Prenatal antibiotics exposure and the risk of ASD

In lines 262-263, the authors state that macrolides are the most frequently prescribed antibiotics. The animal studies the authors describe in this section used nonabsorbable antibiotics (neomycin, succinylsulfathiazole). This is highly misleading. The authors may need to revise this section.

2. In acknowledgments, the authors state that the artwork was prepared using freely available BioRender. The free version of BioRender doesn't provide permission for use in publications. The authors may want to provide clarity on this.

3. 252 references for a 12 pages article (excluding figures and tables) seem to be a bit excessive. 
